# Brief communication: Post-wildfire rockfall risk in the Eastern Alps

Sandra Melzner[1], Nurit Shtober-Zisu[2], Oded Katz[3], Lea Wittenberg[4]

[1]Geoconsult ZT GmbH, Hölzlstraße 5, Wals bei Salzburg, 5071, Austria
[2]Department of Israel Studies, University of Haifa, Abba Khoushy Ave 199, Haifa, 3498838, Israel
[3]Geological Survey of Israel, 32 Yeshayahu Leibowitz St., Jerusalem 9692100, Israel
[4]Department of Geography and Environmental Studies, University of Haifa, Abba Khoushy Ave 199, Haifa, 3498838, Israel

*Correspondence to*: Sandra Melzner (sandra.melzner@geoconsult.eu)

**Abstract.**

In the Eastern Alps, no previous research focused on the impact of wildfires on the occurrence of rockfalls. The investigation of wildfires and post-wildfire rockfalls gains new importance with respect to changes in weather extremes and rapid social developments such as population growths and tourism. The present work describes a wildfire that occurred in August 2018 in a famous world-heritage site in Austria. Indicators of fire severity and rockfall occurrence during and after the fire are described.

**1 Introduction.**

Many areas in the Eastern Alps are prone to rockfalls endangering settlements and infrastructures, causing every year several fatalities. In recent years, wildfires in the Alps and their impact on the environment gain new importance with respect to climate change and rapid social developments such as population growth and tourism.

Most research on the impact of wildfires has been made in the USA and the Mediterranean-climate region (Cerdà, 1998; Cerdà 20   and Doerr, 2005). Although post-wildfire risk from debris flows have been studied by various authors ( Marxer et al., 1998; Conedera et al., 2003; Calcaterra et al., 2005; Cannon et al., 2010;;  Santi et al., 2013), rockfalls associated with wildfires have been poorly studied ( Swanson, 1981; De Graff and Gallegos, 2012; Santi et al., 2013; De Graff et al., 2015;). De Graff et al., (2015) showed that out of sixteen wildfires in California (USA), seven wildfire- affected areas experienced significant rockfall occurrence days after the burn. The slope steepness and underlying lithology were given, showing maximum sizes from 0.30-25   1.85m in their largest dimension and an average of 0.5m. Furthermore, all rockfalls were generated from steep slopes (over 39°) of metasedimentary or granitic lithology experiencing moderate to high soil burn severity.

According to Keeley (2009) the term "fire intensity" is defined as the energy output from fire, whereas the terms "fire severity" and "burn severity" are used interchangeably for the aboveground and belowground organic matter consumption from fire. The term "ecosystem response" is defined as the functional processes that are altered by fire including regeneration, 30   recolonization by plants and animals and watershed. According to Keely (2009) 'soil burn severity" is often used interchangeably with "fire severity". In the USA, it is the preferred term (applied to soils) used in post-fire Burned Area

Emergency Response assessments (Parsons et al., 2010). 'Fire severity', however, is a more comprehensive term that includes also reference to 'vegetation burn severity'.

The aim of this work is to describe the of a wildfire, which occurred in August 2018 at a steep rockwall in the heavily toured world-heritage site "Hallstatt" in the Salzkammergut region in Upper Austria (47°33'27.00" N, 13°38'37.03' E). In order to assess the impact of the wildfire on the recent and future rockfall activity in the area. A helicopter flight and field survey was carried out. The survey was conducted by Sandra Melzner, Geological Survey of Austria, as part of the project "Georisks Austria" (GEORIOS). The focus of the inspection was on the identification of possibly changed rockfall potentials and loosening of the rock due to the strong heat effect (Melzner, 2018) with regards to the rockfall hazard analysis conducted in 2014 (Melzner, 2015). A revisit of the area was conducted in May 2019 to record the temporal post-wildfire changes to the ecosystem.

## 2 Wildfire affected site

### 2.1 Area settings

The wildfire site is situated on the southwest exposed rockwalls of a glacially over steepened Alpine trough valley. The valley is characterised by almost vertical rock walls several hundred metres high, which are mainly made of Mesozoic limestone (Dachstein formation). The limestone is characterized by predominantly thick-bedding, sudden changes in the joint mass structures and the presence of dominant fault systems. In the wildfire affected part of the rockwall, the bedding has a predominantly medium steep dipping of 35 to 45 degrees in the direction of the rockwall (from NE to NW) (Melzner, 2015; 2018). The bedding planes form preferably locations for trees to grow and are usually covered with a thin layer of debris (Fig. 1). A fixed- rope climbing tour is installed in the rockwall, which is frequently used by numerous climbing tourists. The talus slope below the rockwall is relatively short and has an inclination between 30 and 40 degrees. Soil type consists of scree (Ø < ~10 cm), or medium compact soil with small rock fragments and some larger blocks. The scree is covered by a very thin layer of soil and organic matter which can be classified as brown rendzina and brown earth (Fig. 2). Vegetation is characterized by coniferous trees, mainly spruces and broad-leaved trees such as beech and larches. Pre-fire vegetation was composed of medium old forest and an understory of sparse bushes on the rockfall talus slope. The forest on the talus slope beneath the rockwall is designated as protection forest for the houses in the valley floor. Annual precipitation is about 1,743 mm. The highest 1-day precipitation amount since 1901 was measured on August 12, 1959 with 118 mm, and the maximum annual precipitation measured was 2,085 mm (1954). There are 20 to 30 convective summer thunderstorm days per year. Precipitation as snow occurs normally between November to April during which snow cover can reach thicknesses up to a few meters.

## 2.1 Event description

On August 21, 2018 at 09:30 a.m. a wildfire was presumably initiated by a carelessly discarded cigarette or the reflection of a broken glass bottle at the foot of the rockwall. At that time there were three groups of about 20 climbers on the via ferrata. Since the fire could only be extinguished from the air by helicopters, the via ferrata had to be evacuated to protect the climbers from falling rocks and branches caused by the downwind of the helicopters during the fire-fighting work. The fire rapidly spread up the rockwall (area size of about 3 ha) affecting the trees growing mainly on the bedding planes of the limestone (Fig. 1). The protection forest beneath the rockwall was not affected by the fire (Fig. 2). The night from August, 21 to August, 22 the first evacuations of the houses beneath the rockwall took place as burned trunks, rootstocks and rockblocks were falling down the rockwall, the latter approaching two houses. Sixteen mapped rockfall boulders, which reached the settlement area, had volumes smaller 0,3m³ (Fig. 3). With the exception of a minor damage to one building, no severe damages to buildings or injuries of inhabitants occurred. In the following days the firefighter brigades tried to extinguish the fire from above the rockwall with fire hoses and from the air with helicopters carrying water containers/buckets. In total, four police and military helicopters were flying throughout the days, constantly every two minutes refilling the buckets with water from the nearby Lake Hallstatt. During the four days of firefighting operation, up to 100 people (firefighter brigade, police, military, mountain rescue) were on duty every day. Unusual low wind conditions and rainfall (starting on August 24, 2018) prevented the spread of the fire towards the village of Hallstatt. The official end of the firefighting mission was on August, 28 2018. Rockfall hazard and risk assessment conducted by the Geological Survey of Austria (Melzner,2015) formed an important part of the wildfire emergency response. Preventive rockfall hazard actions by the Austrian Torrent and Avalanche Control (WLV) after the wildfire included, (i) establishment of temporary rockfall protection measures (embarkments, simple rockfall fences) in order to be able to clear the wildfire area, (ii) clearance of the wildfire area (removal of loose stones, boulders, trees at risk of falling, etc.), (iii) repair of pre-existing rockfall protective structures damaged by rockfall, and (iv) sowing of seeds in the wildfire affected scree and soil.

## 3 Fire severity measures

### 3.1 Loss and decomposition of organic matter

Indicators of fire severity (Fig. 1 and 2) are the colour of the trees and the decomposition degree of the leafs and needles. Unaffected trees have a green and unaltered colour, whereas burned or heated trees are easily recognizable by the brown colour. Varying degrees of consumption of the needles/leafs and organic matter can be related to different classes of fire severity. According the classification of Keeley (2009), the trees in the affected area show moderate or severe surface burn. This is visible that most of the burned trees still have needles, but all understorey plants and pre-fire soil organic layer (besides a post-wildfire needle cover) are consumed. In the transition area of burned and not-burned area, the vegetation shows indicators for light fire severity, expressed by green needles although the stems may be scorched, and the understory plants and soil organic

layer is largely intact. At the foot of the rockwall we observed a burning tree that has fallen down the rockwall carrying a large rock, which burst into various rockfall boulders during the first impact with the ground.

## 3.2 Changes in soil and rockmass structure

The vertical relief of the rockwalls, the anabatic winds and patchy vegetation pattern, caused an upward jumping of the fire, resulting in a spotty fire pattern (Fig. 1). Thus, the residence time of the fire and the heating duration were reduced, leading to a less direct influence of the high temperatures on the rockmass structure. Fire- induced rock surface alteration and cracking due to thermal shock are typical rock weathering processes occurring during a wildfire (Dorn, 2003; Shtober-Zisu et al., 2015). Thermal shock takes place when the thermally- induced stress event is of sufficient magnitude that the material is unable to adjust quickly enough to accommodate the required deformation and accordingly fails (Hall 1999). As a result, surface failure takes the form of cracking or exfoliation due to the compression and the shear stress it induces (Yatsu 1988). Moreover, rocks composed of several minerals, each with different coefficients of thermal expansion, may experience stresses resulting from the minerals' differential thermal response to heating and cooling cycles (McFadden et al. 2005).

Spalling or the formation of exfoliation fissures (caused by insolation weathering), may be less severe in such exposed terrain conditions than in more gentle slopes (Blackwelder, 1927; Zimmerman et al., 1994; Shakesby and Doerr, 2006; Shtober-Zisu et al., 2015) . In the course of the wildfire, abundant small rock fragments had come to rest directly at the base of the rockwall. The rockfall boulders which got detached of the rockwall during the wildfire could be easily identified in the field, as they usually have at least one black (scorched) side (Fig. 2). Some smaller rockfall boulders with volumes <0.3m³ have reached the valley floor (Fig. 3).

The slope in the uppermost part of the rockwall is covered by gravel, stones and blocks with a matrix composed of fine clastic material and ash. It could be mobilised in the form of a debris slide/flow in a heavy precipitation event. Such an event has not been documented to far in this area. As the organic material mantling the scree slope in the upper part of the rockwall was consumed completely (Fig. 1 and 2), we observed that the ash covers the surface. Ash has a kind of "sealing effect" reducing the infiltration, accelerating the splashing effect and increase the surface runoff (Brook and Wittenberg, 2016). It can be assumed that future frost and thaw cycles will further weaken the rock or that the loose slope debris in the upper rockwall area will be remobilised by heavy precipitation events. In forests, wildfire usually generates a mosaic of different levels of burn severity (Neary et al., 2005). In sites affected by fire of light-to-moderate severity, needle cast occurs when leaves from the scorched trees fall down and blanked the surface, thus protecting the soil from further erosion (Cerdà and Doerr, 2008; Robichaud et al., 2013).There are numerous studies addressing the effect of ash deposits on runoff and erosion processes, rates, and quality (Bodí et al., 2011). Results, however, are inconclusive; while many suggests that ash temporarily reduces infiltration, either by clogging soil pores or by forming a surface crust (Onda et al., 2008),other indicate, that ash and specifically the black char produced during light-to-moderate fires might increase infiltration by storing rainfall and protecting the underlying soil from sealing (Wittenberg, 2012). The ash layers may also protect the burned soil against raindrop impact

and related splash erosion, and its leachates may reduce soil erodibility by promoting flocculation of the dispersed clays (Woods and Balfour, 2008). Ash particles penetrate, accumulate and shelter under the rock spalls formed during the fire, even for several decades (Shtober-Zisu et al., 2018).

## 4 Post-wildfire rockfall risk

An increased rockfall activity of rather smaller rockblocks during but as well after the wildfire is recognizable. The destabilization of small rockblocks and the burn of tree roots may as well cause destabilization of larger rock masses (Fig. 1). These would pose a significant risk to the houses and infrastructures. Above the steep rockwall, some greater boulders in/on top of the scree slope can be remobilized as secondary rockfalls by falling trees or undercutting erosional processes (Fig. 4). The wildfire probably had a superficial impact on the rockmass structure of the vertical rockwalls. According to Thomaz and Doerr, (2014) moderate temperatures (< 400∘C) had the most major effect on soil chemical properties. The study was conducted using a set of thermocouple that were placed at 0-2 cm soil depth. Even relatively low temperatures at the surface of the soil can trigger mineralogical changes.

The burned roots in the joints and profound fractures accelerate physical weathering processes. Chemical weathering of rocks will speed their eventual transformation into secondary clay minerals causing slope instability due to its being a lower strength material than the unweathered rock. The swelling potential of these secondary clays induce significant vertical overpressure, thus reinforcing subsequent progressive rockfall failure.

According to Bierman and Gillespie (1991), wildfires increase rock's susceptibility to weathering through several mechanisms: (1) uneven heating and thermal expansion, along with the vaporization of endolithic moisture, induces spalling; (2) intense heating increases the rate of thermal diffusion significantly and accelerate loss of gases such as Argon, Helium, and Neon from the rock; (3) heating causes microfracturing of rock and could cause the loss of Chlorine-rich fluid from inclusions. Additionally, if the temperatures reached during the burning are high enough, decarbonation in the limestone may occur, enhancing decomposition and further erosion. If calcrete overtop the rock surface, its laminar structure substantially decreases the rocks' tensile strength and the threshold magnitude of the thermal stress needed to weather them. Thus, the laminar structure of the calcrete plays a key role in all types of physical weathering, specifically in the exfoliation process that occurs along the bedding planes between the laminae. The development of empirical relationships for predicting location, magnitude and frequency of increased post-wildfire rockfall activity require further research and collecting more data. Although the mechanism of direct and indirect impact of wildfire on debris flows has been studies in numerous past studies, knowledge about post-wildfire rockfalls is limited and is completely absent in the Alpine region. The observations in the present study imply that falling trees and burned roots might have significant impact on rockfall occurrence during and after a wildfire event, but this issue requires further investigation. Rockfalls during the fire may be triggered by human activities such firefighter actions or winds caused by helicopters during firefighting operations.

The vegetation recovery plays an important role in mitigating post-fire dynamics and increasing land stability. Rates and patterns of post-fire vegetation regeneration were extensively studies in the Mediterranean, however, the Alpine vegetation has gained relatively little attention (Camac et al., 2013). In Austria, a study that documented patterns of post-fire land recovery, indicated that 60 years after a fire trees covered only 40% of the burned area, whilst grassland and exposed rock/debris areas have expanded and remained active. Moreover, it was suggested that the slope will not reach its former

condition before 2070. This extreme window of disturbance of more than 120 years is attributed to the steepness of the slope and to the shallow soils and dolomitic bedrock that were severely damaged by the fire (Malowerschnig and Sass, 2014).

## 5 Conclusions and recommendations

In the Eastern Alps, no work on wildfires and post-wildfire rockfall activity has been published so far. The "August 2018 Hallstatt wildfire" shows clearly, that wildfires can have a significant impact on ecosystems and pose a high risk to settlements

in the Alpine area. Wildfires in steep Alpine valleys behave differently than those on flat or moderate inclined slope. The vertical rockwalls, the anabatic winds and patchy vegetation pattern, caused an upward jumping of the fire resulting in a spotty fire pattern. This most probably results in spatially varying fire intensities, and consequently highly heterogenic changes in soil and rockmass structure. It makes it very difficult to predict future rockfall occurrence and estimate the associated risk. The rockfall hazard and risk assessment conducted in 2014 enabled a fast decision making as part of an emergency response during

and after the wildfire catastrophe in terms of identification of possibly endangered houses but as well of planning of preliminary rockfall preventive measures.

Future research activities should focus on the study of wildfire behaviour in Alpine Valleys. A national wildfire database in combination with a forest inventory map would help to plan forest management strategies for wildfires in the Alpine region. The development of tools to identify the days of high wildfire risk supported by the meteorological survey would enable a fire

hazard rating system. Despite the logistical difficulties in the highly exposed relief, there is a practical need to understand the wildfire induced rock surface alteration and cracking due to thermal shock in order to improve prediction of potential post-fire rockfall problems and associated hazards and risks. The compound impact of fire and snow cover on future rockfall and debris slide/flow activity would be a very important future research topic.

## 6 Acknowledgements

The authors would like to thank the editor Mr. Mario Parise, the reviewer Mr. J.V. De Graff and a second anonymous reviewer for the constructive comments and suggestions to the manuscript.

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

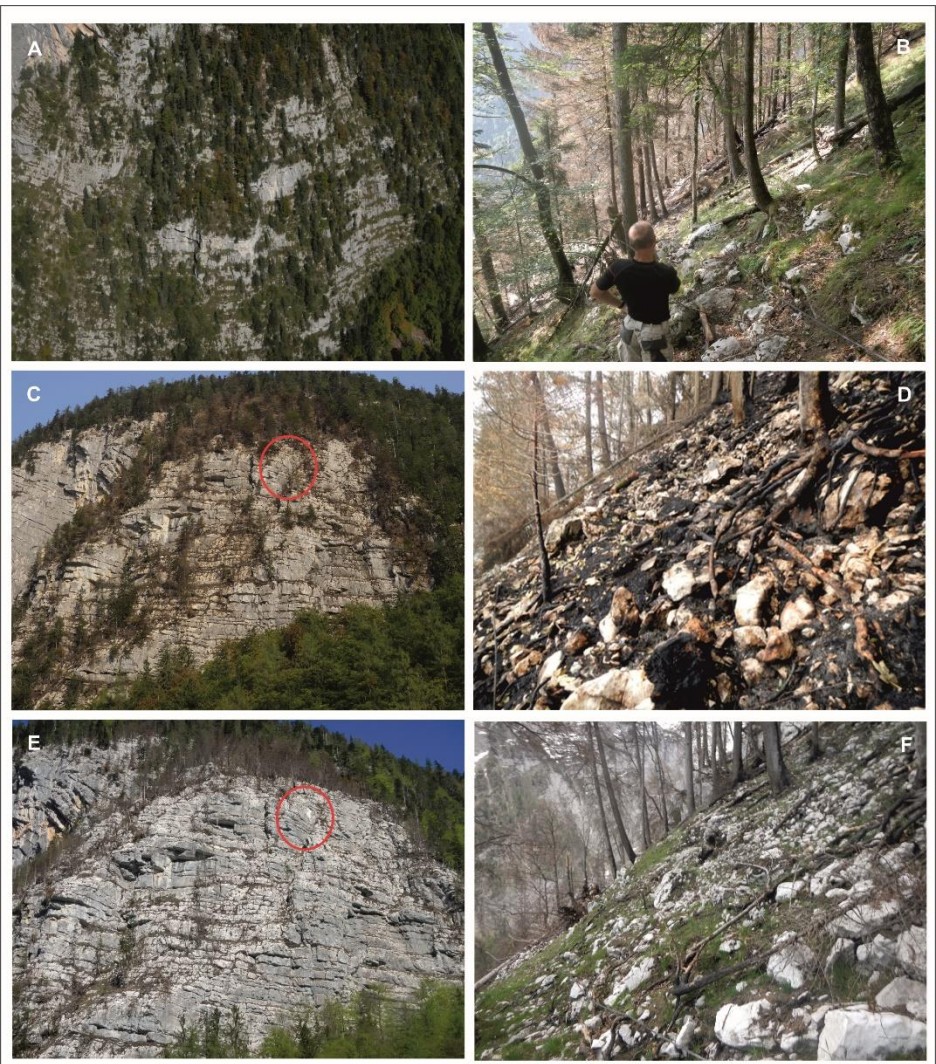

**Figure 1: Temporal changes of the wildfire affected area: biomass, soil and rock characteristics before the wildfire in 2014 (A), in the border between burned and not-burned forest in August 2018 (B), directly after the wildfire in August 2018 (C and D) and eight months after the burn in April 2019 (E and F). Post-wildfire rockfalls (red circle in E) with a volume of about a few m³ is recognizable.**

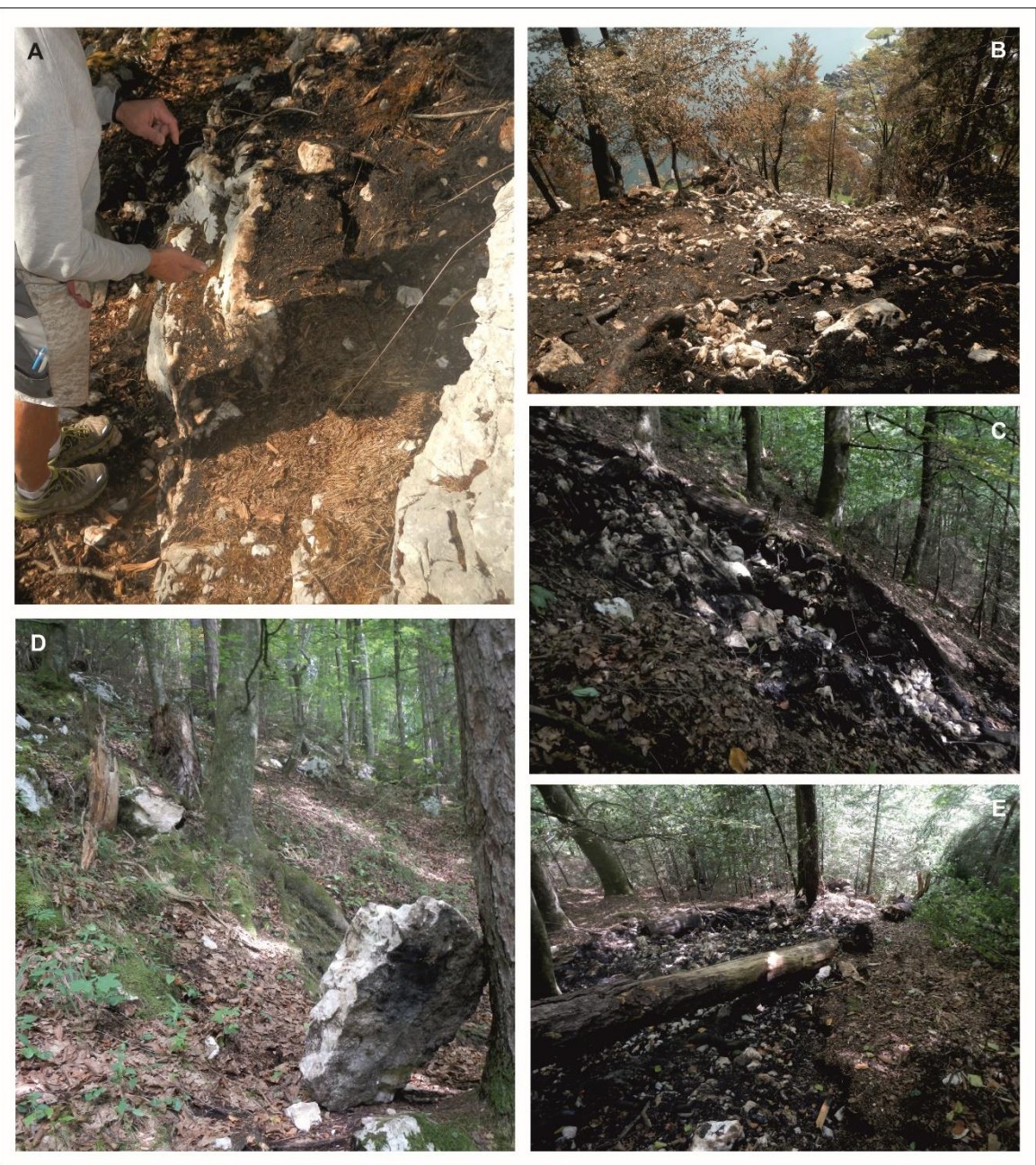

**Figure 2: Indication for wildfire severity. (A) Development of cracks in the scree, (B) ash and needle covers the terrain have a "sealing effect" reducing the infiltration capacity, (C) Talus slope beneath the rockwall is covered by a thin soil layer, (D) rockfall boulders detached during the wildfire are often easily identifiable on the black colour, (E) the protection forest beneath the rockwall only got affected to a very minor extent due to the anabatic winds and high moisture of the brown rendzina/ brown earth.**

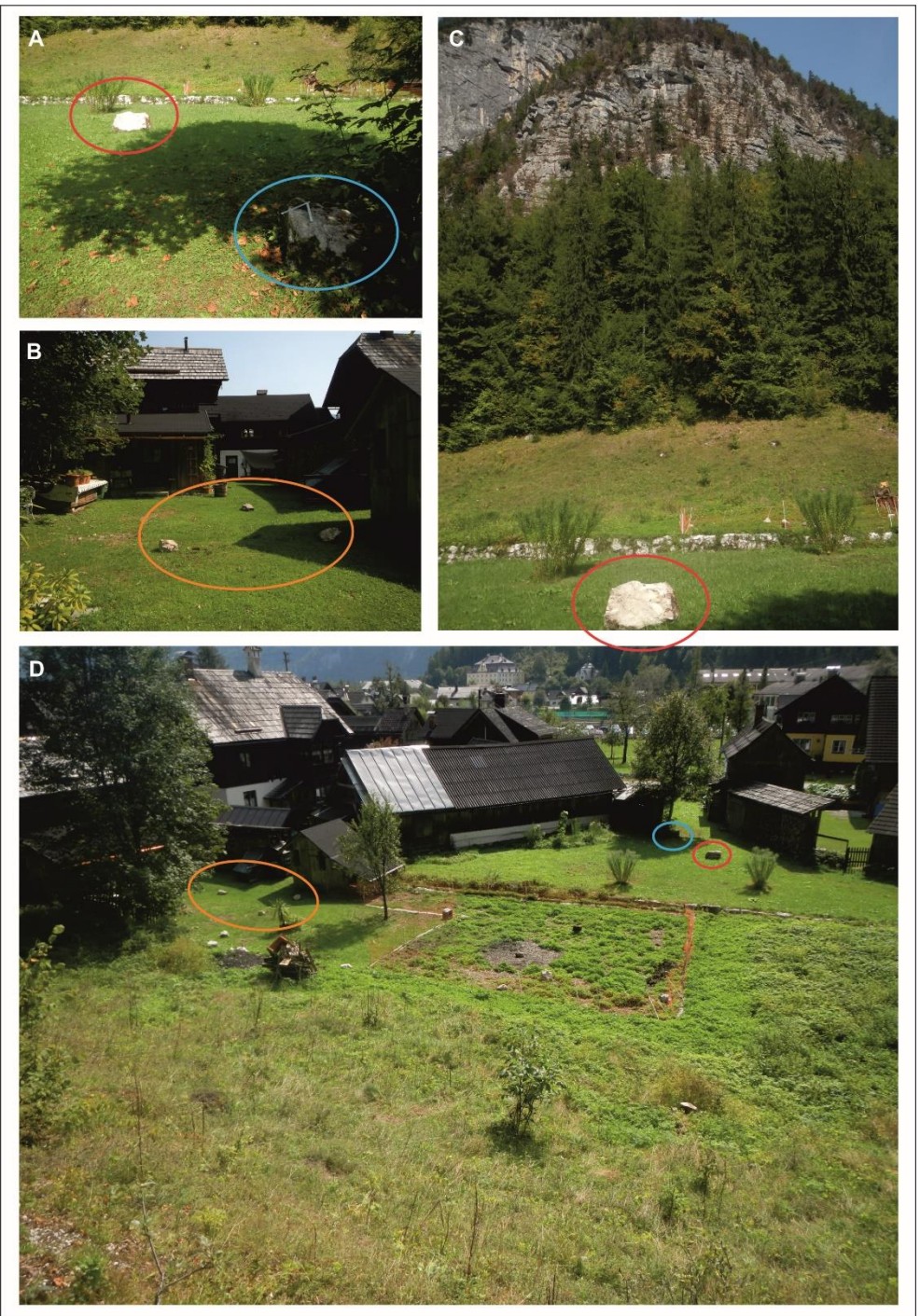

**Figure 3: Rockfall boulders, which were detached during the wildfire and reached the settlement area. An older rockfall boulder (blue circle) marks the maximum reach of past rockfalls and has a volume of about 0,2m³ (0,8*0,6*0,4), the fresh boulder (red circle) a volume of about 0,3m³ (0,8*0.75*0,5).**

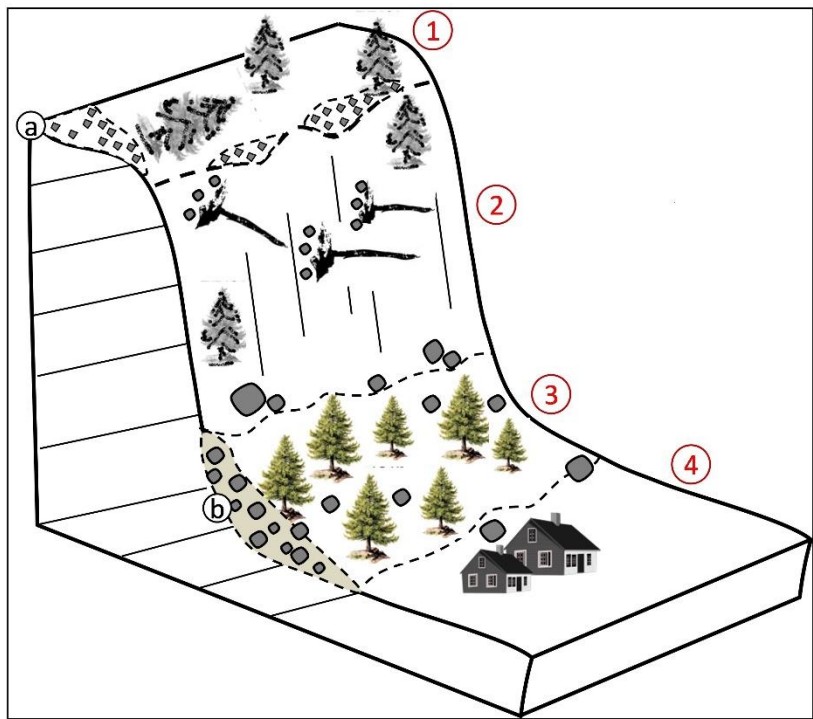

**Figure 4: Sketch of the wildfire affected area. During the wildfire, the trees in the less inclined upper rockwall (1) and in the vertical rockwall (2) show indicators for medium fire severity; the protection forest (3) did not get affected by the wildfire. During the fire, rockfalls were detached from the rockface and by falling trees, which reached the houses in the valley floor (4); (a) scree; (b) talus**
