# Peer review of "Brief communication: Post-wildfire rockfall risk in the Eastern Alps"

_Natural Hazards and Earth System Sciences, 2019_

## Referee Comment (RC1) · Jerome De Graff (Referee) · 12 Aug 2019

General Comments:

The authors have done a good job of describing a particular natural hazard (rockfall after wildfire) in an area where this phenomenon has not previously been described. They further note correctly that this is not a well-studied phenomenon based on available literature. The authors use a good technical approach by examining a recent wildfire affecting a rock wall and its adjacent steep slopes known to be a source of rockfall. The concern is for the potential increase in rockfall in a location where people reside, and tourists regularly come to visit. This forms a case study to examine whether and/or how rockfall risk was influenced by the wildfire's impact to the vegetation and ex-

posed bedrock where rockfall might be initiated. The physical setting, the nature of the wildfire event, and the rockfall during and after the wildfire are well-described. Their work documents increased rockfall during and after the wildfire. Some observations suggest that falling trees and burned roots attributable to the wildfire are important to rockfall occurrence. Quantifying this effect, establishing a fire hazard rating for this region, and collecting additional data on post-wildfire rockfall activity are among the useful recommendations offered based on the Hallstatt wildfire example.

Specific Comments:

The authors define burn severity and fire severity interchangeably as suggested by Keeley (2009), one of their references. The field assessment teams responsible for post-fire hazards evaluation commonly use the term soil burn severity (as the authors note in their description of research described in De Graff et al., 2015). The different levels of soil burn severity (low, moderate and high) are more rigorously defined based on: Parsons, A., Robichaud, P. Lewis, S., Napper, C. and Clark, J (2010) Field guide for mapping post-fire soil burn severity, General Technical Report RMRS-GTR-243. USDA Forest Service, Rocky Mountain Research Center, Fort Collins, CO (USA), 49 p. (downloadable PDF available at: https://www.fs.fed.us/rm/pubs/rmrs_gtr243.pdf). This would make their interpretations of the fire effects a little more understandable for those who have experience in post-wildfire assessments or lack access to Keeley's paper. On line 24, the author's write that all the rockfalls described by De Graff et al. (2015) "..were initiated from scree slopes,. . ." This was not stated in the cited reference. The slope steepness and underlying lithology were given. On line 127, the author's write about chemical weathering into secondary clay minerals "which can approach the properties of a lubricant causing slope instability." Chemical weathering into secondary clay minerals can cause slope instability but is largely due to its being a lower strength material than the unweathered rock. "Lubrication" is an incorrect technical explanation. On lines 132 & 133, the author's write ". . .knowledge about post-wildfire rockfalls is limited, if not completely absent." The assertion "if not

completely absent." seems at odds with the four articles they cite at the beginning of the paper (lines 20-22). It is an insufficiently studied hazard and it might be correctly characterized as completely absent from studies in the Eastern Alps or some similar geographic designation. On lines 142 & 143, The Author's referred to conducting rock-fall modelling to enable fast decision-making during and after the wildfire in terms of identification of possible endangered houses. This item in the Conclusions and Recommendations section is not supported by the previous sections. There are certainly pertinent observations made during that time. But there is no description of this data being provided in real-time to those responsible for public safety or fire suppression efforts. Perhaps, that needs greater elaboration in the earlier sections. Also, I would characterize this as risk assessment as part of an emergency response; not modelling. In the text on lines 93 thru 96, three articles are cited, Hall 1999, Yatsu, 1988, and McFadden et al., 2005. These three cited works are not found in the reference list of this manuscript.

Technical Comments:

Line 10: comma needed after "Alps" and before "no"

Line 18: "population growths" should be "population growth"

Line 24: "seven wildfire affected areas" should be "seven wildfire-affected areas"

Line 32: "very touristic" might be replaced with "heavily toured"

Line 33: Delete extra word ("in") before "A helicopter flight. . .."

Line 34: Might be clearer to say the survey was conducted "as a part of the project. . ." rather than "in the frame of the project. . ."

Line 44: "In the wildfire affected areas" should be "In the wildfire-affected areas"

Line 45: The citation (Melzner, 2015, 2017). Should that not be (Melzner, 2015, 2018)?

Line 51: an understory of "sparsely" bushes. . . should be an understory of sparse

bushes

Line 55: suggest change from "Precipitation as snow occurs normally between November and April snow cover can reach thicknesses. . ." to "Precipitation as snow occurs normally between November and April during which snow cover can reach thicknesses. . ."

Line 58: "glassbottle" should be two words

Line 63 thru 72: Dates should be stated in the format "August 24,2018" rather than 28.08.2018 in several places.

Line 65: ". . .injuries of habitants. . ." should be ". . .injuries of inhabitants. . ."

Line 75: ". . .and (vi) sowing of the wildfire affected scree and soil." Sowing what?

Line 84: "and the undestory plants and soil organic. . ." should be "understory"

Line 90: "Fire induced rock surface alteration. . ." should be "Fire-induced"

Line 92: ". . .the thermally induced stress event. . ." should be "thermally-induced"

Line 122: ". . .and the burn of trees/roots. . ." should be "burning of tree roots"

On the figures: I would make the letters (A, B, etc.) in the multiple image figures larger

---

## Referee Comment (RC2) · Anonymous Referee #2 · 25 Aug 2019

General Comments: The paper describe an interesting event occurred in an area prone to rockfalls but not usually affected by wildfire. The descriptions of the wildfire, of the effect on the vegetation and on the soil/subsoil is exhaustive but qualitative. There are no comparison between areas burned and not burned so it is difficult to assess statistically how the fire affected the occurrence of rockfalls. Of course, it is a preliminary report but some quantitative indication about, for example, rainfalls event that caused rockfall should be given. A map showing wildfire limits and rockfall event may also help the reader to understand the dimension of the event. It could include a burn severity rating. Also a comparison to similar slope not affected by wildfire can be an "added value" in the report. All chapter 4 in full of "probably" and "may be" that limit the paper to a discussion or better to a preliminary report and no more. It is also

not clearly exposed if rockfalling take origin from bedrock or from blocks on talus or already available on the slope. In my experience, very important is also the time of persistence of the fire and the quantity of organic fuel already available on the slope. No discussion about this factor is given. Moreover, if the temperature reached during the burning are high enough, also decarbonatation in the limestone may occur, so a discussion on that kind of data could be interesting. All discussion about post fire risk assessment are sharable. Evolution in time of the vegetation of the burned areas could also give suggestion about the evolution in time of the hazard. The manuscript does not represent a substantial contribution to the understanding of natural hazards and their consequences but is a clear description of field observation and general considerations. The scientific and technical approaches are only described, but any data is given so discussion are only general statements. By the way, the paper is presented in a a clear, concise, and well-structured way

Specific comment A rockfall is a fragment of rock (a block) detached by sliding, toppling, or falling, that falls along a vertical or sub-vertical cliff, proceeds down slope by bouncing and flying along ballistic trajectories or by rolling on talus or debris slopes": this is the Varnes definition. In these cases, and in that referred by De Graff, it seems to me that there isn't the moving in the "free air". So, a more precise description of the kinematic is aspected. Probably, is better to tell about rock bouncing rather that rockfalling! About changes in soil and rockmass structure, in my experience, wildfire interest no more than a few centimeters of soil and probably less on bedrock. Moreover, authors says that the duration of the fire was reduced. So, I have great doubts that mineralogical changes took place and probably only very surficial exfoliation could be developed. Then, rockfalling during the fire could also be induced by human activities, like helicopters and firemen's operations: no discussion about this is presented. The last part of this paragraph deals about rainfals, but no information or correlation between rainfalls and rockfalling in the following days/weeks have been described. Nothing to say about the 2 last paragraphs: I agree with all the considerations and all future development about risk assessment and management of post wildfire rockfalls.

---

## Author Comment (AC1) · 23 Sep 2019

Dear Editor,

Please find our responses, point by point, in red letters. Line numbers refer to the corrected version.

Comments of Referee 1: J.V. de Graff

General Comments:

The authors have done a good job of describing a particular natural hazard (rockfall after wildfire) in an area where this phenomenon has not previously been described. They further note correctly that this is not a well-studied phenomenon based on available literature. The authors use a good technical approach by examining a recent wildfire affecting a rock wall and its adjacent steep slopes known to be a source of rockfall.

Thank you

The concern is for the potential increase in rockfall in a location where people reside, and tourists regularly come to visit. This forms a case study to examine whether and/or how rockfall risk was influenced by the wildfire's impact to the vegetation and exposed bedrock where rockfall might be initiated. The physical setting, the nature of the wildfire event, and the rockfall during and after the wildfire are well-described. Their work documents increased rockfall during and after the wildfire. Some observations suggest that falling trees and burned roots attributable to the wildfire are important to rockfall occurrence. Quantifying this effect, establishing a fire hazard rating for this region, and collecting additional data on post-wildfire rockfall activity are among the useful recommendations offered based on the Hallstatt wildfire example.

Specific Comments:

The authors define burn severity and fire severity interchangeably as suggested by Keeley (2009), one of their references. The field assessment teams responsible for post-fire hazards evaluation commonly use the term soil burn severity (as the authors note in their description of research described in De Graff et al., 2015). The different levels of soil burn severity (low, moderate and high) are more rigorously defined based on: Parsons, A., Robichaud, P. Lewis, S., Napper, C. and Clark, J (2010) Field guide for mapping post-fire soil burn severity, General Technical Report RMRS-GTR-243. USDA Forest Service, Rocky Mountain Research Center, Fort Collins, CO (USA), 49 p. (downloadable PDF available at: https://www.fs.fed.us/rm/pubs/rmrs_gtr243.pdf). This would make their interpretations of the fire effects a little more understandable for those who have experience in post-wildfire assessments or lack access to Keeley's paper.

Manuscript adapted (Please see Lines 30-33; including Parsons et al., 2010)

On line 24, the author's write that all the rockfalls described by De Graff et al. (2015) "..were initiated from scree slopes,. . ." This was not stated in the cited reference. The slope steepness and underlying lithology were given.

Corrected (Line 24)

On line 127, the author's write about chemical weathering into secondary clay minerals "which can approach the properties of a lubricant causing slope instability." Chemical weathering into secondary clay minerals can cause slope instability but is largely due to its being a lower strength material than the unweathered rock. "Lubrication" is an incorrect technical explanation.

Corrected (Lines 136-137)

On lines 132 & 133, the author's write ". . .knowledge about post-wildfire rockfalls is limited, if not completely absent." The assertion "if not C2 completely absent." seems at odds with the four articles they cite at the beginning of the paper (lines 20-22). It is an insufficiently studied hazard and it might be correctly characterized as completely absent from studies in the Eastern Alps or some similar geographic designation.

Corrected to "…and is completely absent in the Alpine region". (Line 153).

On lines 142 & 143, The Author's referred to conducting rockfall modelling to enable fast decision-making during and after the wildfire in terms of identification of possible endangered houses. This item in the Conclusions and Recommendations section is not supported by the previous sections. There are certainly pertinent observations made during that time. But there is no description of this data being provided in real-time to those responsible for public safety or fire suppression efforts. Perhaps, that needs greater elaboration in the earlier sections. Also, I would characterize this as risk assessment as part of an emergency response; not modelling.

Corrected accordingly. (Lines 171-172)

In the text on lines 93 thru 96, three articles are cited, Hall 1999, Yatsu, 1988, and McFadden et al., 2005. These three cited works are not found in the reference list of this manuscript.

Added: Lines 214, 225, 250.

Technical Comments:

Authors agreed to all technical comments and adapted the manuscript accordingly

Line 10: comma needed after "Alps" and before "no" Line 18: "population growths" should be "population growth" Line 24: "seven wildfire affected areas" should be "seven wildfire-affected areas" Line 32: "very touristic" might be replaced with "heavily toured" Line 33: Delete extra word ("in") before "A helicopter flight. . .." Line 34: Might be clearer to say the survey was conducted "as a part of the project. . ." rather than "in the frame of the project. . ." Line 44: "In the wildfire affected areas" should be "In the wildfire-affected areas" Line45: The citation(Melzner,2015,2017). Should that not be(Melzner,2015,2018)? Line 51: an understory of "sparsely" bushes. . . should be an understory of sparse C3

bushes Line55: suggest change from "Precipitation as snow occurs normally between November and April snow cover can reach thicknesses. . ." to "Precipitation as snow occurs normally between November and April during which snow cover can reach thicknesses. . ." Line 58: "glassbottle" should be two words Line 63 thru 72: Dates should be stated in the format "August 24,2018" rather than 28.08.2018 in several places. Line 65: ". . .injuries of habitants. . ." should be ". . .injuries of inhabitants. . ." Line 75: ". . .and (vi) sowing of the wildfire affected scree and soil." Sowing what? Line 84: "and the undestory plants and soil organic. . ." should be "understory" Line 90: "Fire induced rock surface alteration. . ." should be "Fire-induced" Line 92: ". . .the thermally induced stress event. . ." should be "thermally-induced" Line 122: ". . .and the burn of trees/roots. . ." should be "burning of tree roots" On the figures: I would make the letters (A, B, etc.) in the multiple image figures larger

We would like to thank Mr. De Graff for the very constructive comments and helpful suggestions.

---

## Author Comment (AC2) · 23 Sep 2019

Dear Editor,

Please find our responses, point by point, in red letters. Line numbers refer to the corrected version.

Comments of Referee 2

GeneralComments:

The paper describe an interesting event occurred in an area prone to rockfalls but not usually affected by wildfire. The descriptions of the wildfire, of the effect on the vegetation and on the soil/subsoil is exhaustive but qualitative.

There are no comparison between areas burned and not burned so it is difficult to assess statistically how the fire affected the occurrence of rockfalls. Of course, it is a preliminary report but some quantitative indication about, for example, rainfalls event that caused rockfall should be given.

To apply statistical methods to rockfalls is very difficult, because the data on rockfalls is often very limited and thus not statistical representative. Most statistical methods can't cope with little data sets (Melzner et al. in prep.). In the present work "a brief communication" was chosen, because not a lot of data and experience about wildfire induced rockfalls is existent. The scope of this brief communication is to stress the need of research and publish preliminary results. Rainfall is important, as one (among many other) parameters affecting rockfall occurrence and surely within the scope of future research.

A map showing wildfire limits and rockfall event may also help the reader to understand the dimension of the event.

In figure 1 the wildfire and rockfall affected area in the rockwall is displayed (3 ha) and coordinates of the study site are given. Unfortunately, the manuscript is already longer than most brief communications accepted by the journal. We will surely add another map in a future, full research paper.

It could include a burn severity rating. Also a comparison to similar slope not affected by wildfire can be an "added value" in the report.

We address the severity in lines 88-92 but in a very limited form, due to the restrictions of a "brief communication"

All chapter 4 in full of "probably" and "may be" that limit the paper to a discussion or better to a preliminary report and no more. It is also not clearly exposed if rockfalling take origin from bedrock or from blocks on talus or already available on the slope.

The origin of rockfalls is both from rockwalls and rockfalls (from scree) are affecting the settlement area. Authors agree on comment and erased some "probably" and "may be".

In my experience, very important is also the time of persistence of the fire and the quantity of organic fuel already available on the slope. No discussion about this factor is given.

Time of persistence of fire is mentioned in the paper in terms of "spotty fire pattern" in lines 96 and 169. The organic matter accumulated on the slopes is mentioned in lines 54 and 89-92.

Moreover, if the temperature reached during the burning are high enough, also decarbonatation in the limestone may occur, so a discussion on that kind of data could be interesting.

We added a discussion on this topic. Lines 140-149.

All discussion about post fire risk assessment are sharable. Evolution in time of the vegetation of the burned areas could also give suggestion about the evolution in time of the hazard.

We added a discussion on this topic. Lines 157-164.

The manuscript does not represent a substantial contribution to the understanding of natural hazards and their consequences but is a clear description of field observation and general considerations. The scientific and technical approaches are only described, but any data is given so discussion are only general statements.

We agree with this comment, however the scope of this brief communication is to present first observations on an uncommon wildfire event in the Alps and to stress the extremely needed further research on this topic – especially in a zone that experienced very few past wildfires. Further research is surely needed to understand such evens and their consequences, that in recent years become more and more frequent

By the way, the paper is presented in a a clear, concise, and well-structured way

Thank you

Specific comment

A rockfall is a fragment of rock (a block) detached by sliding, toppling, or falling, that falls along a vertical or sub-vertical cliff, proceeds down slope by bouncing and flying along ballistic trajectories or by rolling on talus or debris slopes": this is the Varnes definition. In these cases, and in that referred by De Graff, it seems to me that there isn't the moving in the "free air". So, a more precise description of the kinematic is aspected. Probably, is better to tell about rock bouncing rather that rockfalling!

Vertical rockwalls simplify the "falling through the air" (see fig. 1 and 4). The kinematic description will be further referred in a future full research paper on this topic. Unfortunately, the "brief communication" cannot include a longer introductory chapter.

About changes in soil and rockmass structure, in my experience, wildfire interest no more than a few centimeters of soil and probably less on bedrock. Moreover, authors says that the duration of the fire was reduced. So,I have great doubts that mineralogical changes took place and probably only very surficial exfoliation could be developed.

We added a discussion on this topic. (Lines 132-135)

Then, rockfalling during the fire could also be induced by human activities, like helicopters and firemen's operations: no discussion about this is presented.

We added a sentence on this topic. (Line 155)

The last part of this paragraph deals about rainfals, but no information or correlation between rainfalls and rockfalling in the following days/weeks have been described.

Rainfall might be an extremely important agent inducing rockfalls. However , this issue is not under the scope of the current paper .

Nothing to say about the 2 last paragraphs: I agree with all the considerations and all future development about risk assessment and management of post wildfire rockfalls.

The authors would like to thank the reviewer for the very useful comments that helped improving this manuscript and will surely be taken into consideration in future research in this topic.

---

## Editor Decision (ED1)

[revised manuscript text omitted]